# A Pilot Study to Test the Feasibility of a Home Mobility Monitoring System in Community-Dwelling Older Adults

**DOI:** 10.3390/ijerph16091512

**Published:** 2019-04-29

**Authors:** Heesook Son, Hyerang Kim

**Affiliations:** Red Cross College of Nursing, Chung-Ang University, 84 Heukseok-ro, Dongjak-gu, Seoul 06974, Korea; hson@cau.ac.kr

**Keywords:** mobility, home monitoring system, older adults living alone, visiting nursing, technology

## Abstract

Technology enables home-based personalized care through continuous, automated, real-time monitoring of a participant’s health condition and remote communication between health care providers and participants. Technology has been implemented in a variety of nursing practices. However, little is known about the use of home mobility monitoring systems in visiting nursing practice. Therefore, the current study tested the feasibility of a home mobility monitoring system as a supportive tool for monitoring daily activities in community-dwelling older adults. Daily mobility data were collected for 15 months via home-based mobility monitoring sensors among eight older adults living alone. Indoor sensor outputs were categorized into sleeping, indoor activities, and going out. Atypical patterns were identified with reference to baseline activity. Daily indoor activities were clearly differentiated by sensor outputs and discriminated atypical activity patterns. During the year of monitoring, a health-related issue was identified in a participant. Our findings indicate the feasibility of a home mobility monitoring system for remote, continuous, and automated assessment of a participant’s health-related mobility patterns. Such a system could be used as a supportive tool to detect and intervene in the case of problematic health issues.

## 1. Introduction

In Korea, individuals aged 65 years or older exceeded 14% of the population and entered into an aged society in 2017 [1]. It is expected that Korea will become a super-aged society in which more than 21% of the population is aged 65 years or older within a decade, while this population is estimated to represent approximately 40% of the total population in 2060 [2]. As the number of older adults has increased rapidly, the proportion of older adults living alone has also increased, which was estimated as 7.4% in 2013 and is projected to double (15.4%) by 2035 in Korea [2]. It has been reported that older adults who live alone have a higher vulnerability to adverse health outcomes, including poor physical and psychosocial health conditions, unhealthy lifestyle, and low health care utilization [3,4]. As compared to those living with other family members, older adults living alone are 1.7 times more likely to encounter premature death independent of their health status and other social and functional statuses [5]. The key challenge for the healthcare of older adults living alone lies in the lack of informal caregivers, especially family caregivers, who can support usual physical and psychosocial needs and even respond to emergencies. This lack increases unmet support needs of such older adults and delays access to healthcare services [6]. Although older adults living alone may experience social isolation and lack of support, as well as more frequently encounter substantial and/or potential health risks and limited health resources, many prefer to live in a familiar environment, in which they feel comfortable [7]. 

Increases in the number of older adults, particularly those living alone, and the preference for independent living in one’s own home have triggered the need for modifications to the healthcare paradigm in terms of shifting the focus of the healthcare sector from the hospital/clinic to the home and community. This transformation requires the provision of proactive and personalized care to enable older adults to live independently, safely, and healthily in their home [8]. This has been conceptualized as “P4 medicine”, as follows: identifying health risks by remote and continuous tracking of biological, behavioral, and environmental factors (predictive and preventive); customizing plans for health management from diagnosis to rehabilitation for different health conditions based on individuals’ unique genetic, medical, and environmental characteristics (personalized); and facilitating patients’ engagement in personal health management (participatory) [9].

Advances in health care technologies play an important role in the transition of health care and realization of P4 medicine, to support chronic conditions with more continuous, higher quality of care at home. Various advanced technologies such as wearable devices, embedded sensors, and ubiquitous networking have been utilized to improve monitoring of individuals and communication with health care providers [10]. These tools enable continuous monitoring of individuals’ health conditions with a variety of indicators, which have been used in health care management to guide health care providers toward more effective clinical diagnoses, interventions, and treatments. In response to the need for transformation of nursing practices, technology has been also applied in a variety of nursing processes, including nursing administration, interdisciplinary communication, clinical decision support, and access to medical information. 

Older adults may experience substantial health-related decreases in their functional performance of daily activities that directly influence their health status and quality of life [9]. Therefore, it is important to assess mobility during their daily life so as to provide critical insight into any changes in their functional status and health outcomes. Sensor technology enables automation of the assessment of mobility patterns in an unobtrusive and real-time manner while individuals perform daily routine activities [11,12]. In addition, sensor technology plays a part as a decision support tool, allowing health care providers to remotely detect changes in health-related mobility patterns and to proactively address the causes of these changes [13]. While there are previous investigations of the effectiveness and feasibility of home mobility monitoring systems in individuals, little is known about the best way to integrate these systems into visiting nursing practice.

In Korea, three types of home care nursing services are provided: hospital-based home care services, community-based visiting nursing, and long-term care insurance-based visiting nursing. Hospital-based home care services were initiated in 2000 according to the Medical Act. When individuals who need long-term care are discharged from a hospital, medical care services are prescribed by the physicians. These services include medication, injection, laboratory tests and collection of specimens, therapeutic nursing, and education. Community-based visiting nursing provides rehabilitative and preventive care, as well as basic medical care, health education, and referral services to people with chronic diseases or disabilities in the community according to the Regional Public Health Act established in 1994. This service is provided free of charge to socially vulnerable individuals, that is, older adults, refugees, or those who have low income [14]. Long-term care insurance-based visiting nursing was established in 2008 to support the healthy lives of community-dwelling older adults aged 65 years or older or those aged less than 65 years but who have degenerative chronic diseases such as dementia and cerebrovascular disease. These home care services include home care nursing, general assistance, baths, day care, and short stays. Both community-based and long-term care insurance-based visiting nursing are provided by public health nurses working at public health centers. A total of 2,242 visiting nurses are currently employed at community health centers nationwide and, on average, each nurse visits approximately 570 households per year [15]. Since more frequent visits are needed for individuals with serious or multiple health problems, the number of visits that a nurse makes needs to be greater. Therefore, the shortage of personnel and increased burden on the nurses are important issues for the public health centers. The shortage of community or public health nurses and burden on the public health system are not problems unique to Korea [16,17]. However, the steep increase in the aging population and growth in the number of people with long-term health conditions are more profound in Korea than in other countries, and the best ways to support community-based visiting nursing services need to be established.

The purpose of this study was to test the feasibility of a home mobility monitoring system among older adults living alone in the community. As a pilot study, we tested the feasibility using both quantitative and qualitative methodological approaches. The sensor outputs were analyzed quantitatively. To assess adaptability of the system, interviews were conducted with the participants, while brief survey questions were distributed to visiting nurses. The study results would provide practical evidence regarding continuing and automated monitoring of daily activities for older adults living alone and insight to guide successful adoption and implementation of the system in visiting nursing practice.

## 2. Materials and Methods 

### 2.1. Participants

The following inclusion criteria were used: adults aged 65 years or older who were living alone and who had no limitations or impairments in independent functioning regarding activities of daily living. After obtaining approval from the Institutional Review Board (IRB No: 1041089-201606-HRSB-115-01CCCC) of the institution that the principle investigator was affiliated to, participants were recruited from two community health centers in Korea, using a convenience sampling method. Research nurses approached older adults who were visiting the community health center and provided them with information, including the purpose of the study and eligibility (i.e., participants needed to be living alone). After obtaining written consent form the older adults who voluntarily agreed to participate in the study, the nurses visited the participants. To ensure understanding of the purpose and procedures of the study, the staff showed the participants mock devices used for mobility monitoring, after which informed consent was obtained. Between the start of recruitment in October 2016 and April 2017, mobility monitoring sensors were installed in eight older adults’ households and maintained in those locations for 15 months. For the first three months, the sensor data were used for the stabilization period to increase accuracy and validation of the sensor outputs (these data were excluded from the analysis). Therefore, only data collected during the 12 months after sensor stabilization were included in the analysis.

### 2.2. Data Collection Via the Home Mobility Monitoring System 

The mobility monitoring system used in this study consisted of five components (Hidea Solutions Co., Ltd, Seoul, Korea): activity sensors, base unit (Gateway), door sensor, flame and gas detector, and help trigger. The activity sensors produce the sensor output periodically (every 2 s) within a 4 m radius. These sensors were installed on the ceiling of various locations in the house, including the bedroom, living room, kitchen, and bathroom (Figure 1). The number of sensors installed depended on the size of the space and number of rooms. The base unit works on mobile networks and transmits sensor output to a data server. The door sensor was attached to the main entrance and detected entry and exit. The help trigger was an emergency call button by which a resident could call 119 (the nationwide emergency telephone number in Korea) at the time of an emergency (i.e., a fall).

During the study period, participants were asked to perform their habitual activities as usual. The total daily sensor outputs were recorded with time stamps, calculated for each daily indoor activity, and presented as the activity index (AIx). As a dimensionless number, AIx indicates the total amount of indoor activities, derived from the calculation of the frequency of detection for each period of time and the duration of data transmission [18]. Mobility data were monitored by trained research nurses. To identify any unusual patterns in relation to health conditions, the research nurses reviewed the sensor outputs on a daily basis throughout the study period. To ensure the accuracy and validation of the sensor outputs during the first 14 days, after a 3-month system stabilization period, the research nurse made a phone call to the participants and asked about their daily activities using daily activity questionnaires. These questionnaires included the following information: “What time did you sleep last night?”; “What time did you wake up today?”; “How would you rate your sleep quality last night?” (good/moderate/bad)”; “What time did you eat breakfast/lunch/dinner yesterday?”; “Did you go out at all yesterday? (yes/no); If ‘yes,’ When? (from 00:00 to 23:59)”; “How would you rate your health status yesterday? (good/moderate/bad).” Then, an hourly timetable of daily activities was created based on the questionnaire and classified into three categories: sleeping, indoor activities, and going out. If the sensor outputs did not correspond to the activity questionnaires, further investigations were conducted to examine whether the sensor was operating correctly (i.e., the sensor company was contacted). 

### 2.3. Data Analysis 

The sensor outputs over the initial 14 days of the 1-year study period after exclusion of the first 3 months were used to establish the baseline daily activity pattern. This was categorized into three classes of activities to compare with activity patterns determined by the self-reported questionnaire: Sleeping was determined based on wake-up time (point of time showing significantly increased sensor outputs); sleep-onset time (point of time showing significantly decreased sensor outputs); and time of going out was determined based on a lack of sensor outputs between two consecutive time points after the door sensor was detected. Everything else was categorized as indoor activities time. If the classes of activities measured by sensor outputs during the 14-day baseline period were not comparable to the classes of activity patterns reported in the daily activity questionnaire, the day was considered as potentially exceptional. The maximum, minimum, and mean ± 2 SDs of the sensor outputs for each class of activity were calculated after excluding the days with exceptional activity patterns. The reference values for atypical activity patterns were determined as mean ± 2 SDs of total daily AIx during the baseline period [19]. Based on the analysis of the sensor outputs, the mobility pattern was classified into typical or atypical. The automated alarm system sent alarm messages to the research nurses when the mobility pattern deviated from the established usual pattern. When an unusual pattern was found, the research nurse called the participant and investigated the reasons for the atypical activity pattern (i.e., bad health condition, having visitors, or long-time absence from home, etc.). 

### 2.4. Evaluation of Adaptability 

After the data collection was completed, face-to-face interviews were conducted with the participants to assess the adaptability of this system and to learn their impressions of the system. Each interview was conducted at the participant’s house for approximately 30–60 min. They were asked to provide various types of feedback, including their general satisfaction, any concerns and expectations regarding system use, intentions to use such a system in the future, and any suggestions of how to improve the system. Their views of the usefulness of the system were also queried, for example, to determine what features of the system were helpful to them. In addition, we distributed brief survey questions to visiting nurses affiliated to two community health centers used for the participant recruitment, and thereby obtained a needs assessment of the home mobility monitoring system. After identifying the priority of tasks for which they were responsible, the nurses were queried regarding expectations and intentions to use the system for each task. The mean scores of expectations and intentions to use were calculated. 

## 3. Results

### 3.1. Identification of Daily Indoor Activity

Usual daily activities obtained by questionnaire were comparable to activity patterns determined by sensor outputs. Types of housing and lifestyles, including the time points of sleep-onset and wake-up, activity level, and pattern varied among participants (Table 1). The average sensor outputs over the selected 14-day baseline period were determined for each class of daily indoor activity including sleeping, indoor activities, and going out (Figure 2). Sensor outputs during sleep showed lower activity than other classes of activities. Each class of daily indoor activity was clearly differentiated by sensor outputs.

### 3.2. Determination of Atypical Patterns

All sensor outputs during the 14-day baseline period of all participants were within the reference value for each class of daily activity pattern, except for one day for one participant, which was identified as atypical, as the sensor outputs for that day deviated from the reference value and thereby generated an alarm. For example, Figure 3 shows sensor outputs on a day: expressed as hourly means + 2 SDs during the initial 14-day period of typical activity pattern (A); on a day of typical activity pattern during the baseline period (B); and for an atypical activity pattern (C). According to the self-administered daily activity questionnaire, participant E (female/81-year-old) usually slept at 10:00 pm and woke up at 6:00 am. This was comparable to the means + 2 SDs of AIx collected during the initial 14-day baseline period (A). However, the last graph (C) shows that sensor outputs from 1:00 am through 3:00 am were higher compared to the reference value for the 14-day baseline period. The activity level was higher than at any time point on other days of the baseline period. On other days around that day, the sensor outputs during sleep were normal (comparable to B), which was within the means of baseline (A). The research nurse called the participant the next day and asked about the participant’s health condition and what happened overnight. It was reported that the participant did not sleep well and went to the bathroom frequently because of diarrhea during the night. 

The other case of the health-related mobility problem was identified. For participant C (female/93-year-old), AIx during the daytime gradually decreased and dropped below the baseline minimum value from November 10 to 17, 2016 (Figure 4). The research nurse called the participant and identified that she had become too frail to engage in daily routine activities. Therefore, the research nurse called the participant’s daughter to ask her to visit. In response to the request by the research nurse, the daughter’s family visited and took the participant to the hospital for treatment. The visitors generated the increased sensor output between the initial decline of the participant and hospitalization (date range 21 November 2016–25 November 2016). 

### 3.3. Adaptability of the System to Participants: Satisfaction and Perceived Usefulness 

To evaluate the adaptability of the system in use, the participants were asked about their satisfaction and whether they perceived the system was useful. They noted general satisfaction with the system. One participant said: “I feel good to have this. I feel secured. If something bad happens to me, it can recognize and connect someone to help me because I live alone. My oldest son is currently living in a foreign country and another is living in Seoul, but he does not frequently come to see me because he is busy. Both he and his wife are working and children are too young. But having this, I feel more secured than before.” Another participant said: “When my neighbor came to my house and saw the sensor, she wondered what it was. I told her that it saves me and she said it was good for me. I think so.” All respondents reported that they were not bothered by the system. It was noted that “When it was installed in my room at first, I was concerned about privacy matters, but I was insensitive to that so soon.” Only a participant wore the necklace of emergency pager. A participant noted that “I always keep it… even when going out. I place it at the bedside during sleep… to use it in case. It is so dependable.”

Regarding the usefulness of the system, the participants suggested it would be most helpful for someone who was unable to move independently or who had more a serious medical condition, especially dementia or stroke. It was considered particularly helpful for persons such as who live alone or have no-one to help them. When participants were asked about the intention to keep on the system, they generally agreed they would do so. One participant said that “I don’t care. Because it doesn’t bother me”, while another stated, “I am still young to use it. I am healthy and have no limitations to act freely. I don’t need help yet. But when I get older I would need something like that, just in case.” However, some participants were worried about the cost. One participant said: “If it is expensive for maintenance, I don’t want to have it. I have the intention to use it if it is at the cost of the current electronic fee.” 

### 3.4. Adaptability of the System Use from Visiting Nurses: Expectations of Usefulness

The tasks ranked as highest priority were, in order from most to least important, general management of the registered clients for home-visit services, home-visit practice for provision of basic care (including regular physical examinations and health education), and home-visit practice for individuals at high risk (including intensive care for older adults with chronic disease and those living alone). They were also involved in project planning and evaluation, group education, health campaigns, and administrative tasks. The visiting nurses highly rated the usefulness of the home mobility monitoring system for the management of individuals at high-risk (mean rating: 7.8 out of 10 points). The next most highly rated was the system’s utility for general management of registered individuals (7.7/10), followed by utility for home visits for basic care services (6.5/10). The system was expected to support decision-making processes (6.8/10), increase work efficiency (6.6/10), shorten working hours (5.5/10), and reduce workload (5.1/10).

## 4. Discussion

We tested the feasibility of a supportive tool for community-based visiting nursing services, namely a home-based mobility monitoring system for continuous, automated, and real-time monitoring of daily activity patterns in community-dwelling older adults. Our study revealed that an individual’s mobility features, including total amount of indoor activities, sleep patterns, and frequency of bathroom visits, discriminated and clearly differentiated atypical patterns even over a year, which validated the use of sensor technology to monitor mobility. The system was also considered feasible and adaptable from perspectives of both the participants and visiting nurses. These observations suggest that the technology has potential benefits for older adults living alone in a free-living environment.

Several challenges arose during this study. First, the system was only available to measure indoor activity so that activities outside the home and total daily activities could not be estimated. However, the duration and frequency of going out can support prediction of daily activity patterns. For example, a person with mobility limitations may go out less frequently, stay outside for shorter periods, or stay inside less actively [20]. Second, there were technical challenges to discriminate data from multiple persons (e.g., visitors or pets). For example, sometimes total sensor outputs in a day were higher than other usual days. This was because the participant had visitors on that date. Different activities of participants and visitors were simultaneously collected via the same sensors and accumulated in a data server. 

Wearable sensors may be used as an alternative to address these issues, by providing total daily indoor and outdoor activity and enabling the tracking of target individuals. However, discomfort of on-body wearable sensors is a problem. These are also less adaptive to older adults even in a free-living setting [21,22]. Additional effort in terms of education and training is needed to use these devices and their accessories. Furthermore, regularly recharging the battery and wearing the devices distracts individuals from habitual daily activities. Finally, as the devices are bothersome to use, this consequently decreases satisfaction and adherence to using the devices [23,24]. Thus, accuracy and reliability of data cannot be guaranteed in this population [25,26]. In the current study, we did not ask the participants to use a wearable device, although we provided an emergency pager for them to carry. However, the participants were reluctant to carry this pager due to its inconvenience, which replicates the limited adaptability of wearable devices for this population. 

Another alternative technology can be used to determine the presence of visitors. For example, visitor-counting sensors at the door can generate signals while a person is passing [27]. Visitor-counting sensors can work in various ways, including by visual appearance of a visitor, heat emission, reflections of the body surface, or pressure against the floor [28]. Additional technical components would be helpful for enhancing functionality of the system to discriminate extrapersonal or environmental factors (e.g., visitors). However, in terms of digitization of healthcare, health professionals should maintain balance between productivity or effectiveness and ethical concerns (e.g., safety and security, privacy, maintaining a trusting relationship with patients, and fair and equitable access to quality of services) [29]. Although health monitoring technology has benefits, it may limit interactions with patients for contextual understanding of individual situations and the patient’s participation in decision-making [30]. In particular, the interaction between health professionals and health service users, both patients and healthy individuals, is a type of social support. Thus, communicating by digital interface with older adults living alone may be not the best option for ensuring quality of care. Health professionals should consider whether it is appropriate to apply health information technology and it is effective for patients as a decision-support tool. 

Our study also has limitations that restrict generalization of the results. This study was a small pilot study with insufficient duration of follow-up, and included only healthy individuals without limitations on daily activities imposed by health problems. However, despite the small sample size, the follow-up period was longer than in previous studies and identified health-related mobility patterns. Previous studies primarily used performance-based episodic measurements [31,32], at specific time points [33], or in an experimental environment [34,35]. Previous studies performed tests over relatively short-term periods in small populations and thus failed to capture changes in health-related mobility patterns over time [36,37,38]. Cavanaugh et al. [39] and Raknim et al. [40] conducted experiments by which to quantify patterns of physical activities over a year in a free-living environment, using wearable sensors and a smartphone application respectively. Recently, investigations have been conducted of the feasibility and validity of using ambient sensors for mobility monitoring of older adults. Stack et al. [41] tested the usefulness of home-based mobility monitoring sensors to determine fall risks among five older adults with Parkinson’s disease, for six weeks at their own home. However, the purpose of the study was to determine the movement patterns over a short period that signify higher risk of falling, rather than to monitor time-dependent changes in movement patterns and thereby infer health status. Similar to the current study, Suzuki et al. initially tested the feasibility of a monitoring system similar to our system to identify mobility patterns over a six-month period in a single older adult living alone [42]. The researchers then conducted similar work to determine mobility patterns over four weeks in relation to health status in three older adults living in a nursing home. The installation of sensors was limited to the participants’ own room, which were only available for public spaces of nursing home [18]. The system successfully identified patterns of daily indoor activities but did not detect changes in health-related mobility patterns due to the short duration of the experiment. Compared to previous studies, the current empirical data were collected with a larger sample size, for longer period of time, and supported the adoption of home-based mobility monitoring technology in real practice as a potentially powerful methodology for continuous, real-time assessment and proactive intervention to prevent adverse health outcomes. 

Another limitation of our study is that we did not assess biopsychosocial characteristics of the participants. Considering that impaired mobility in later life has been recognized as an early predictor of impairment of physical and psychosocial function, recognition of multifaceted interactions among physical, psychological, social, and behavioral factors would be helpful to understand contextual changes in mobility patterns and to provide customized solutions for health management. Although this feasibility study focused on how well the system functioned to identify indoor mobility patterns of older adults during their daily lives and to determine physical conditions from mobility patterns, we suggest that assessment of biopsychosocial factors associated with physical activity would provide a more detailed view of habitual mobility patterns. For example, the International Classification of Functioning, Disability, and Health (ICFDH) enables comprehensive description of functional status along biological, psychological, social, and environmental dimensions, thus presenting health and disability as a single spectrum [43]. Prospective studies for longer periods of time would provide rich information on the changes in mobility patterns that imply a particular health status and permit more practical insight into health management for healthy aging in place. 

Finally, we assessed nurses’ general expectations regarding home mobility monitoring systems in visiting nursing practices, rather than their pragmatic acceptance of the current system in particular. Further study would be required to practically determine the acceptability of this system to visiting nurses, which is another determinant of successful implementation of technology in nursing care services. With the increasing need for technological support for a variety of nursing services, advances in technology will provide many opportunities to facilitate nursing care processes: enabling informed decision-making and managing demand for nursing care services; and improving access to and sharing of health information via websites and other technology-mediated applications that relay information [44,45]. Consequently, technology may improve the quality of care and enhance the working conditions of nurses. Nevertheless, despite these advantages, the use of technology in nursing practice is not always successful in terms of the work efficiency and satisfaction of nurses, since implementation of technology requires changes in habitual work processes [46]. In particular, technology use requires changes in the workflow, clinical process, and care delivery [47,48]. As a result, technology may produce a greater burden on nurses. For example, poor use of technology may increase the workloads of nurses, which may serve as a barrier to uptake of technology. Ammenwerth et al. [49] noted that the time spent planning and documenting tasks was longer with a computerized system. Daly et al. [48] found that the use of technology did not significantly improve nursing practices and work conditions. In this study, the nurses did not choose to use the computerized system to make decisions related to the diagnosis more frequently than the conventional approach. Inappropriate technology applications, such as poor technology design and interface, may result in lower usability and acceptability by nurses [49]. Finally, nurses’ perceptions, attitudes, and motivations toward the technology play crucial roles in its successful application [48]. In real-world nursing practice, inappropriate implementation of technology can impede its success [48]. Therefore, the specific needs for technology and its impact on the nursing care workflow should be considered in technology adoption. The use of technology should be guided by explicit protocols aligned with nursing care processes. Appropriate implementation of technology based on well-established nursing protocols with a specific purpose may promote successful technology implementation and use.

Despite limitations, this study has meaningful implications for the use of technology for unobtrusive, automated, remote, and real-time monitoring of vulnerable populations such as older adults living alone in the community. This preliminary result could lead to clinical applicability in terms of preventive and proactive interventions using technology in community nursing practice. For example, in the case of participant C, the sensor output markedly increased after an initial decline below the baseline value. It was determined that the daughter’s family stayed at the participant’s home during that period to take care of the participant. An increase in the number of persons in the house led to an increase in the cumulative sensor outputs. Although the situation was not fully characterized (i.e., how many visitors there were and for how long they stayed), the sensor output proximal to the event was consistent with the brief report by the daughter. Although cumulative sensor outputs engender complexity and confusion when interpreting the target individual’s mobility, the outputs provide meaningful information regarding situational context in an unobtrusive manner. The functionality of automated alarm generation suggests another significant benefit of applying the system to older adults living alone since the sensors capture unusual patterns of activity and inform visiting nurses and/or their family. Strict establishment of baseline activity patterns is essential to ensure accuracy and reliability of alarm signals. Daily activity patterns of older people are known to be relatively stable [49]. To evaluate changes in health-related functional mobility, therefore, it is necessary to establish usual daily activity patterns while an individual is healthy or stable, namely before he or she encounters illness or disability. 

Aging is accompanied by multiple chronic diseases and complex health-related issues, which influences physical, social, and emotional independency and quality of life in the older adult population [50]. From a biopsychosocial perspective, mobility, and particularly functional ability/disability, is influenced by a broad range of factors, including physical, cognitive, psychosocial, environmental, and economic [9,51]. Thus, mobility is widely accepted as a fundamental determinant of active aging and independency, and is closely associated with health outcomes and quality of life in later adulthood [51,52]. There is a rapidly emerging variety of mobility-monitoring devices that can define and measure health-related mobility. These emphasize the comprehensive assessment and monitoring of mobility to capture changes in health status. It is important to detect functional decline of older adults in timely manner so that they can maintain their daily activity levels. This approach can promote healthy aging and improve quality of life. The use of technology for health care provides many opportunities not only to enable management of health care service demand and decision making for health care providers, but also to more actively engage in monitoring a patient’s condition [53]. With the use of this technology, visiting nurses can identify health changes earlier and provide proactive care by monitoring patients’ information on an ongoing basis without making daily home visits. This might contribute to reducing emergency department visits, admissions, and readmissions, playing a supportive role in promoting response to the onset of preventable chronic health problems and emergencies such as loss of consciousness and falling [54].

## 5. Conclusions

To address the need for home-based nursing care services, the current study tested the feasibility of a home mobility monitoring system as a supportive tool for community-based visiting nursing. The preliminary findings showed that the sensor outputs collected via the mobility monitoring system reflected the participants’ daily activity patterns and discriminated atypical patterns. Follow-up mobility monitoring for the remaining study period may provide further information on participants’ health status, which will be used as evidence for developing a visiting nursing protocol. It is expected that the findings of this study will contribute to addressing health-related mobility problems among older adults using technology in an unobtrusive, automated, and continuous manner in a free-living environment. This study also provides preliminary evidence to help establish guidelines for the implementation of technology-based community nursing services.

## Figures and Tables

**Figure 1 ijerph-16-01512-f001:**
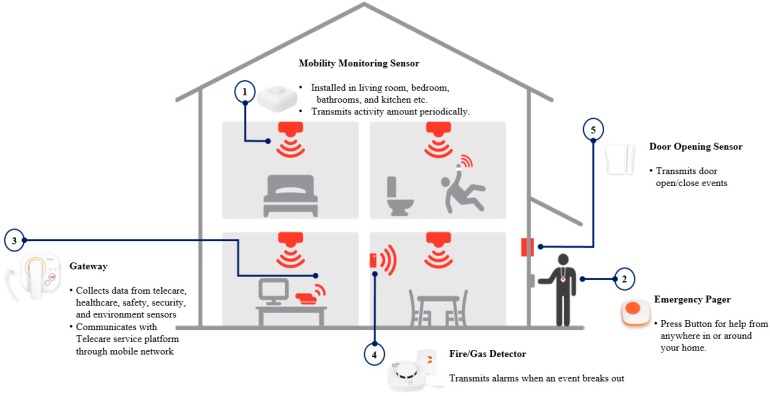
General features of the home mobility monitoring system.

**Figure 2 ijerph-16-01512-f002:**
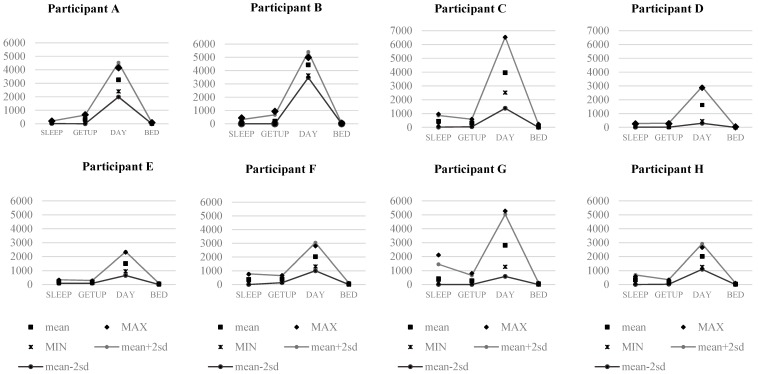
Patterns of daily indoor activities during the initial 14-day baseline period. SLEEP: the average of sensor outputs during sleep; GETUP: the average of sensor outputs at the point of time of wake-up; DAY: the average of sensor outputs during the daytime; and BED: the average of sensor outputs at the point of time of going to bed.

**Figure 3 ijerph-16-01512-f003:**
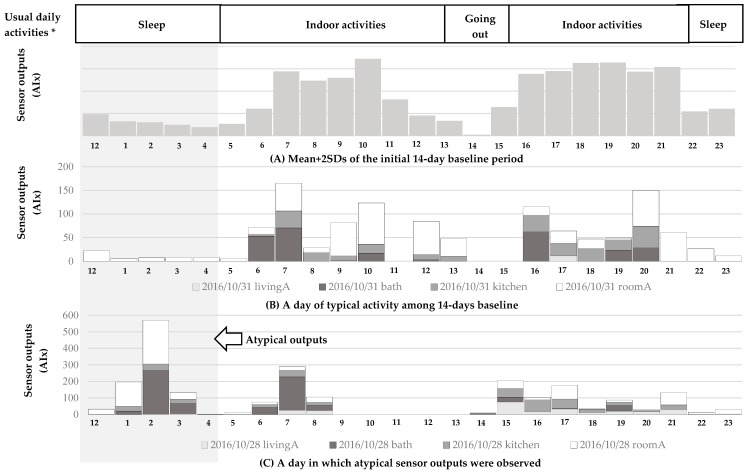
A typical sensor output of Participant E (F/81-year-old).

**Figure 4 ijerph-16-01512-f004:**
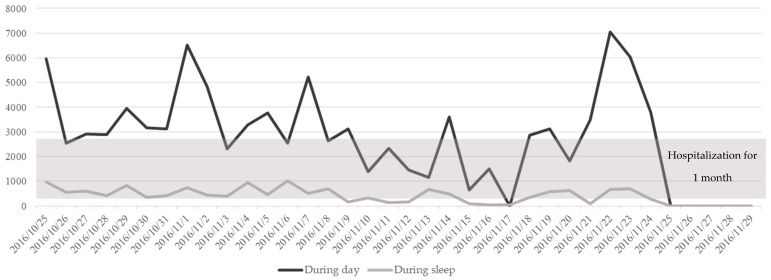
Changes in total indoor activities of Participant C (F/93-year-old).

**Table 1 ijerph-16-01512-t001:** Baseline characteristics of older adult participants.

Participants	A	B	C	D	E	F	G	H
Sex/Age, years	F/87	F/69	F/93	M/89	F/81	F/79	F/74	F/71
Comorbidities	Hypertension Cancer Arthritis	Cancer Asthma	-	-	Diabetes Cancer	Hypertension Diabetes Heart disease	Hypertension	Hypertension Diabetes Cerebrovascular disease Cancer
Housing	Studio apartment	Studio apartment	Apartment	Studio apartment	Apartment	Studio apartment	Studio apartment	Apartment
Number of rooms *	2	2	4	2	4	2	2	3
Location of sensors	Bedroom Bathroom	Bedroom Bathroom	Bedroom Bathroom Living room Kitchen	Bedroom Bathroom	Bedroom Bathroom Living room Kitchen	Bedroom Bathroom	Bedroom Bathroom	Bedroom Bathroom Livingroom (Kitchen)
Bedroom is separate?	No	No	Yes	No	Yes	No	No	Yes
Sleep on the bed?	No	No	Yes	No	Yes	No	No	No
Chair or couch in the living room?	No	No	No	No	Yes	No	No	No
Table and chairs in the kitchen or dining room?	Yes	No	No	No	Yes	No	No	Yes
Washstand is installed in the bathroom?	No	Yes	No	No	Yes	No	No	No
Bedtime	21:00	23:00	21:00	18:00	22:00	21:00	23:00	22:00
Waking time	07:00	05:00	05:00	05:00	06:30	08:00	08:00	08:00
Sleep duration	7 h	5 h	7 h	10 h	8 h	10 h	8 h	9 h
Quality of sleep	Fair	Good	Fair	Fair	Fair	Fair	Good	Fair
Awaken during sleep	Everyday, Not specified times	Not at all	Every day, Not specified times	Every day, 1–2 times/night	Every day, every 2 h	Every day, 1–5 times/night	Sometimes, 1 time/night	Almost every day, 1–3 times
Take a nap	Not at all	Not at all	Everyday	Not at all	Sometimes	Not at all	Not at all	Not at all
Physical condition	Fair	Fair	Good	Good	Fair	Fair	Good	Fair

* included bedroom, bathroom, living room.

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
