# Peer review of "A Pilot Study to Test the Feasibility of a Home Mobility Monitoring System in Community-Dwelling Older Adults"

_ijerph, 2019, doi:10.3390/ijerph16091512_

Round 1
Reviewer 1 Report
This is an interesting study on a system to monitor indoor/home mobility of older persons who live in the community by themselves. In particular, the study demonstrates a way to detect atypical mobility patterns that suggest additional investigation by nurses and possibly interventions. I have the following comments and suggestions for the authors on how to improve the manuscript. 1. Please specify how your convenience sample was actually established. Were older adults who visited community health centers approached by nurses during their visits, was some snowball sampling involved, etc.? 2.The actual duration of the study period from which data are presented here is never stated. There was a 3 months adjustment period, then a 14 day period to establish baseline levels, but how long was observed after this? At some point the authors mention a total study period of one year. In the conclusion however, they mention that "[f]ollow-up mobility monitoring for the remaining study period" is yet going to happen. 3. Why is the participant C's mobility increasing after the initial decline and just before hospitalization? Is this because the mobility of the daughter or some other care giver? 4. In relation to this, also discuss how the issue with the recording of the mobility of pets or visitors in addition to the mobility of the person in question could be dealt with. You rightfully outline problems that may arise when participants have to wear a device. There may however be other solutions such as recording who visits the participants for how long. 5. Eventually, I find that large parts of the discussion introducing the importance of mobility as an indicator for general health and quality of life as well as the advantages of telehealth and home-based monitoring could be moved to the introduction or is a repetition of the introduction. I would rather have the authors provide a summary of their findings, compare their findings with results from similar studies, outline limitations of their study in a more detailed way (e.g. small sample, possibly short duration, lack of data on the perspective of nurses who are supposed to interpret the data), and discuss implications for future research necessary until such system can be upscaled.Author Response
This is an interesting study on a system to monitor indoor/home mobility of older persons who live in the community by themselves. In particular, the study demonstrates a way to detect atypical mobility patterns that suggest additional investigation by nurses and possibly interventions. I have the following comments and suggestions for the authors on how to improve the manuscript.
Point 1: Please specify how your convenience sample was actually established. Were older adults who visited community health centers approached by nurses during their visits, was some snowball sampling involved, etc.?
Response 1:
Thank you for pointing this out. We have tried to clarify the sampling method and revised the sentences in the section of 2.1. Participants as follows: After obtaining approval from the Institutional Review Board (IRB No: 1041089-201606-HRSB-115-01CCCC) of the affiliated institution of the principle investigator, participants were recruited from two community health centers in Korea, using a convenience sampling method. Research nurses approached older adults who were visiting the community health center and provided information including the purpose of the study and eligibility (i.e. living alone) for them. After obtaining written consent form the older adults who voluntarily agreed to participate in the study, the nurses visited the participants (lines 107 – 113).
Point 2: The actual duration of the study period from which data are presented here is never stated. There was a 3 months adjustment period, then a 14 day period to establish baseline levels, but how long was observed after this? At some point the authors mention a total study period of one year. In the conclusion however, they mention that "[f]ollow-up mobility monitoring for the remaining study period" is yet going to happen.
Response 2:
Thank you for the clarification. Whole period of installation of sensors were 15 months. Data collected for the first three months were used for stabilization of the sensor and data collected following the next 12 months were used for analyses in this study. We have revised the sentences to clarify the study procedure regarding data collection (lines 116– 119).
Point 3: Why is the participant C's mobility increasing after the initial decline and just before hospitalization? Is this because the mobility of the daughter or some other care giver? In relation to this, also discuss how the issue with the recording of the mobility of pets or visitors in addition to the mobility of the person in question could be dealt with. You rightfully outline problems that may arise when participants have to wear a device. There may however be other solutions such as recording who visits the participants for how long.
Response 3:
We agree with you that more explanation is needed for the participant C. We have made the sentences to help the readers understand this result. The reason for the increase in the mobility followed by the initial decline was described more in detail (lines 220 – 224). We found that the daughter’s family stayed for a while and then hospitalized the participant. We have also addressed this result in the discussion section (lines 414 – 425).
Point 4: Eventually, I find that large parts of the discussion introducing the importance of mobility as an indicator for general health and quality of life as well as the advantages of telehealth and home-based monitoring could be moved to the introduction or is a repetition of the introduction. I would rather have the authors provide a summary of their findings, compare their findings with results from similar studies, outline limitations of their study in a more detailed way (e.g. small sample, possibly short duration, lack of data on the perspective of nurses who are supposed to interpret the data), and discuss implications for future research necessary until such system can be upscaled.
Response 4:
Thank you for the constructive comment. We have restructured the manuscript to avoid redundancy. We have tried to focus more on providing the strengths and limitations of the study and implications for further study in the discussion session.

Reviewer 2 Report
Thank you for the opportunity to review this interesting paper. Research type are few and the topic in certainly important. However, I have serious concerns regarding the introduction and discussion.
Because of your aim is to test the feasibility of a home mobility monitoring system, I think It is mandatory to add in the introduction (and discussion) the biopsychosocial approach of the ICF (international classification of functioning, disability and health - Wolrd Health Organization) and explain why your research did not refer to this approach and did not collect data referring to the ICF.
Author Response
Point 1: Thank you for the opportunity to review this interesting paper. Research type are few and the topic in certainly important. However, I have serious concerns regarding the introduction and discussion.
Because of your aim is to test the feasibility of a home mobility monitoring system, I think It is mandatory to add in the introduction (and discussion) the biopsychosocial approach of the ICF (international classification of functioning, disability and health - World Health Organization) and explain why your research did not refer to this approach and did not collect data referring to the ICF.
Response 1:
Thank you for pointing this out. We absolutely agree with you that the biopsychosocial approach for the ICF would be useful for the study that tests the feasibility, but we did not use the ICF to assess multifaceted health determinants in perspective of biopsychosocial approach because the purpose of this study focuses more on technical validation of the system to capture the mobility patterns of the individuals in the small sample size. However, we have addressed the approach of the ICF for the mobility monitoring in the discussion section (lines 364-378 and 428-436). In addition, we will definitely consider the application of the ICF in the future work using large sample size for longer period since we believe the ICF provides more comprehensive perspectives on health-related functional mobility, determining differences within and between participants.

Reviewer 3 Report
Dear authors,
I have read your paper with great interest, as I have a background in smart home technologies and their implementation in aged-care myself.
Your participants are healthy older people, at least, they were able to carry out ADLs independently. Therefore, I would not speak of patients in your text but just about older people or participants.
The list of references are up to date and the ethics has been taken care of well. The introduction contains a very rich set of background figures and data, and is well written.
My main concerns are the following. Your work is a small pilot study, which is actually well-described for the things you have done. But you also look at the implementation in visiting nurses' practice. Therefore, I am confused why no qualitative approach has been chosen as well (mixed methods), for instance by interviewing how participants and visiting nurses perceived the technology and how they think it can help them in their daily lives/work patterns. This type if data is lacking, but it is the icing on the cake. It would help to lift this work up, beyond the level of another pilot study, and give meaning to your findings. A system like yours may be fully functional, but it is also relevant to nursing practices, and do older people perceive it like a true support for ageing-in-place?
Without such data, your paper is one of the so many pilot studies of smart home technologies described. In its current form, it would not add much to the existing scientific literature.
Author Response
Point 1: I have read your paper with great interest, as I have a background in smart home technologies and their implementation in aged-care myself.
Your participants are healthy older people, at least, they were able to carry out ADLs independently. Therefore, I would not speak of patients in your text but just about older people or participants.
Response 1:
Thank you for pointing this out. We absolutely agree with you that the individuals who participated in this study were not “patients” since their ADLs were independent. For the study population, we have changed the word of patients to participants or individuals throughout the manuscript.
Point 2: The list of references are up to date and the ethics has been taken care of well. The introduction contains a very rich set of background figures and data, and is well written.
My main concerns are the following. Your work is a small pilot study, which is actually well-described for the things you have done. But you also look at the implementation in visiting nurses' practice. Therefore, I am confused why no qualitative approach has been chosen as well (mixed methods), for instance by interviewing how participants and visiting nurses perceived the technology and how they think it can help them in their daily lives/work patterns. This type if data is lacking, but it is the icing on the cake. It would help to lift this work up, beyond the level of another pilot study, and give meaning to your findings. A system like yours may be fully functional, but it is also relevant to nursing practices, and do older people perceive it like a true support for ageing-in-place?
Without such data, your paper is one of the so many pilot studies of smart home technologies described. In its current form, it would not add much to the existing scientific literature.
Response 2:
Thank you very much for the constructive comment. We definitely agree with you that it would be more meaningful to test the feasibility using both quantitative and qualitative methodological approaches. To assess adaptability of the system, we conducted interviews from the participants and distributed brief survey questions to visiting nurses after the data collection was completed (lines 96-99, 172-186). We thought that it would be better to include only the results from sensor outputs in the current study and to place other findings behind (lines 225-289). However, we believe that it is more critical for “the feasibility” to use both approaches quantitatively and qualitatively. Therefore, we have revised the purpose of this study and included more results from the interviews and survey. In addition, we have addressed the implications of these findings in the discussion section (lines 316-336).

Round 2
Reviewer 2 Report
.
Reviewer 3 Report
Dear authors,
I think this paper is ready to be accepted. I am pleased with your amendments to the original manuscript.
There is one small thing, please remove all the words "elderly" from your text and replace it by older people, older adults, or seniors. It is more age-friendly than the word "elderly".